# Oscillations of As Concentration and Electron-to-Hole Ratio in Si-Doped GaAs Nanowires

**DOI:** 10.3390/nano10050833

**Published:** 2020-04-27

**Authors:** Vladimir G. Dubrovskii, Hadi Hijazi

**Affiliations:** 1Universitetskaya Embankment. 13B, Saint Petersburg State University, 199034 Saint Petersburg, Russia; 2Kronverkskiy prospekt 49, ITMO University, 197101 Saint Petersburg, Russia; hijazi@itmo.ru

**Keywords:** vapor–liquid–solid growth, GaAs nanowires, Si doping, electron-to-hole ratio, oscillations of as concentration

## Abstract

III–V nanowires grown by the vapor–liquid–solid method often show self-regulated oscillations of group V concentration in a catalyst droplet over the monolayer growth cycle. We investigate theoretically how this effect influences the electron-to-hole ratio in Si-doped GaAs nanowires. Several factors influencing the As depletion in the vapor–liquid–solid nanowire growth are considered, including the time-scale separation between the steps of island growth and refill, the “stopping effect” at very low As concentrations, and the maximum As concentration at nucleation and desorption. It is shown that the As depletion effect is stronger for slower nanowire elongation rates and faster for island growth relative to refill. Larger concentration oscillations suppress the electron-to-hole ratio and substantially enhance the tendency for the p-type Si doping of GaAs nanowires, which is a typical picture in molecular beam epitaxy. The oscillations become weaker and may finally disappear in vapor deposition techniques such as hydride vapor phase epitaxy, where the n-type Si doping of GaAs nanowires is more easily achievable.

## 1. Introduction

Semiconductor nanowires (NWs) show great promise as fundamental building blocks for use in nanoscience and nanotechnology [1,2,3]. III–V NWs and heterostructures based on such NWs are interesting for applications in nanophotonic devices, particularly those monolithically integrated with a Si electronic platform [4,5,6,7,8,9,10]. These applications of III–V NWs require a controllable methodology for their n-type and p-type doping. Consequently, significant efforts have been put into the investigation of the NW doping process (see, for example, references [11,12,13] for a review). Most III–V NW are epitaxially grown by the vapor–liquid–solid (VLS) method with metal droplets [14]. Si is routinely used as an n-type dopant of planar GaAs layers in molecular beam epitaxy (MBE) [15], but it often becomes a p-type dopant for VLS GaAs NWs [16,17,18,19,20,21,22]. This amphoteric effect has been attributed to a low As concentration in metal droplets, catalyzing the VLS growth of GaAs NWs [22]. 

Due to a low As content in a catalyst droplet, a fractional monolayer (ML) of GaAs NW rapidly consumes a substantial fraction of as atoms available for growth, after which the droplet must be refilled from vapor. This leads to the so-called nucleation antibunching in NWs [23] and manifests in the periodically changing morphology of the growth interface, which is detectable by the in situ growth monitoring of GaAs NWs inside a transmission electron microscope (TEM) [24,25,26,27]. For very low As content, fractional ML can even stop growing at a certain size and then evolve much slower at a rate of refill [28]. Periodic oscillations of the As concentration over the ML growth cycle further decrease the average electron-to-hole ratio achievable with Si doping of VLS GaAs NWs and therefore increase the tendency for p-type doping. This effect has recently been studied in Reference [29] under the assumption of instantaneous ML growth until reaching the “stopping” size. Instantaneous development of the ML in the initial stage requires that the characteristic time of island growth τ is much shorter than the equivalent deposition time of 1 ML $t_{ML}$, which is inversely proportional to the As deposition rate. Such a time-scale separation is long known in the NW growth modeling [30] and is confirmed by in situ growth monitoring in a TEM [24,25,26,27]. 

However, we note that in situ TEM data are obtained for relatively low NW growth rates, typically less than 0.15 ML/s, which is usual for molecular beam epitaxy (MBE). The growth rates can be much faster in vapor phase deposition techniques, reaching approximately 15 ML/s in the extreme case of Au-catalyzed GaAs NWs in hydride vapor phase epitaxy (HVPE) [31]. Such a huge difference (approximately 100 times) does not guarantee that the strong inequality $\tau /t_{ML}\ll 1$
is satisfied in the entire range of possible VLS growth conditions (separation of the ML nucleation, growth, and refill has recently been investigated in Reference [32] from a different perspective). Here, we investigate theoretically self-regulated oscillations of group V concentration in a catalyst droplet during the VLS growth of III–V NWs and their impact on the Si doping of GaAs NWs in the general case without the time-scale separation of the ML growth and refill. Our considerations should apply equally well to any epitaxy technique, which is not necessarily restricted to slow NW elongation. Several other factors influencing the As depletion are studied, including the “stopping effect” at very low As concentrations, the role of the maximum As concentration at nucleation, and its desorption at elevated growth temperatures. We map out the electron-to-hole ratio in Si-doped GaAs NWs in different regimes with either large concentration oscillations leading to the suppression of n-type doping, or very weak oscillations leading to negligible effect. These results should be useful for better understanding the concentration oscillations under different conditions and controlling the related Si doping process in VLS GaAs NWs. 

## 2. Model

According to Reference [22], the ratio z of the atom fractions of silicon atoms replacing Ga and As atoms in solid GaAs is given by $z=\exp(\mu_{5}-\mu_{3}+A),$ with $\mu_{5}$ and $\mu_{3}$ as the chemical potentials of As (labeled “5”) and Ga (labeled “3”) atoms in a catalyst droplet (in thermal units of $k_{B}T$, with $k_{B}$ as the Boltzmann constant and T as the absolute temperature), and A as a constant which represents the difference of internal energies of a GaAs cell with Si impurity replacing either the As or Ga atom. Replacing Ga with Si atoms creates an As−Si pair and produces electrons, whereas replacing As with Si atoms creates Ga−Si pairs and produces holes. Hence, z=n/p is the ratio of electron-to-hole concentrations (electron-to-hole ratio for brevity). The NW is n-doped at z>1 and p-doped at z<1. The replaced atoms should be put back to liquid, which explains the presence of the corresponding chemical potentials in the equation. The tendency for n-type doping at larger $\mu_{5}-\mu_{3}$ is clearly seen. This observation was used in Reference [22] to explain why adding the Si impurity to VLS GaAs NWs leads to predominantly p-type conductivity in MBE (low $\mu_{5}$) [16,17,18,19,20,21] but yields n-type conductivity in HVPE (high $\mu_{5}$) [22]. 

As discussed in detail in Reference [29], the main contribution to$\;\mu_{5}$ is logarithmic [3,32,33], and it can be put as $\mu_{5}=\ln\left(\theta_{L}/\theta_{eq}\right)$. Here, we present the As concentration in a catalyst droplet in terms of the effective coverage equivalent to the arsenic content in liquid, denoted$\;\theta_{L}$, by normalizing the total number of As atoms in liquid to the number of As atoms (or GaAs pairs) in the full ML of a GaAs NW. In this normalization, the size of fractional ML θ changes from zero to unity, so θ represents the ML coverage. The equilibrium As content under no-growth conditions equals$\;\theta_{eq}$, and it depends on the Ga concentration for Au-catalyzed VLS growth [33]. The As concentration in a catalyst droplet is always much lower than that of Ga [32]; therefore, the chemical potential oscillations over the ML formation cycle are due to oscillations of $\mu_{5}$ at $\mu_{3}≅const$ [23]. For the Si doping of GaAs NWs, these considerations yield
(1)$$\frac{z}{z_{0}}=\frac{\theta_{L}}{\theta_{0}},$$
with$\;z_{0}$ as the maximum electron-to-hole ratio corresponding to the maximum As content at nucleation $\theta_{0}$. The value of $\theta_{0}$ is directly related to the nucleation-mediated NW growth rate for both Ga-catalyzed [34] and Au-catalyzed [35] VLS growths of GaAs NWs. This normalized electron-to-hole ratio oscillates in synchronization with the ML growth cycle. The average electron-to-hole ratio in the whole NW is obtained by the corresponding averaging of Equation (1) over one ML growth cycle. 

The kinetics of the ML coverage and as concentration in liquid is described by [29]
(2)$$\frac{d\theta }{dx}=\frac{\theta_{L}-\theta_{eq}}{\varepsilon },\;\frac{d\theta_{L}}{dx}+\frac{d\theta }{dx}=1-{\left(\frac{\theta_{L}}{\theta_{v}}\right)}^{2}.$$

The initial conditions are given by θ(t=0)=0 and $\theta_{L}\left(t=0\right)=\theta_{0}$. The parameters are defined as
(3)$$x=t/t_{ML},\;\varepsilon =\tau /t_{ML}.$$

Clearly, x is the coverage equivalent to refill, which is proportional to the growth time t, and ε is the ratio of the island growth time τ over the deposition time$\;t_{ML}$. As discussed above, the previous results of Reference [29] correspond to the limiting case of instantaneous growth at ε→0. Equation (2) for θ shows that the fractional ML progresses at the rate ε in this normalization when $\theta_{L}>\theta_{eq}$, and it stops growing when $\theta_{L}=\theta_{eq}$. Equation (2) for $\theta_{L}$ shows that the total number of As atoms in liquid and solid changes with time due to As deposition and desorption at the rate of refill. Quadratic dependence of the desorption term on $\theta_{L}$ is due to the desorption of As in the form of As_2_ molecules [33]. The desorption rate from the droplet is determined by the equivalent as content $\theta_{v}$ corresponding to equilibrium of this droplet with vapor providing a given flux onto the droplet surface. The average electron-to-hole ratio is given by
(4)$$\frac{\langle z\rangle }{z_{0}}=\frac{1}{\theta_{0}\left(x_{g}+x_{r}\right)}\int_{0}^{x_{g}+x_{r}}dx\theta_{L}\left(x\right).$$

Here, $x_{g}$ and $x_{r}$ are the normalized growth times of the ML growth and the refill stages, respectively, which are defined as
(5)$$\theta \left(x_{g}\right)=1,\;\theta_{L}\left(x_{g}+x_{r}\right)=\theta_{0}.$$

Our model contains four plausible parameters: $\theta_{0}$ standing for the initial as content at nucleation,$\;\theta_{eq}$ for its equilibrium content, $\theta_{v}$ for desorption, and ε for the growth kinetics. The parameters $\theta_{0}$, $\theta_{eq}$, and $\theta_{v}$ scale linearly with the corresponding atomic concentrations of As and with the NW radius R, so all θ are proportional to R at the fixed concentrations. $\theta_{v}$ is inversely proportional to the square root of the vapor flux and hence decreases for lower fluxes. Lower $\theta_{v}$ corresponds to higher fractions of re-evaporated As atoms. The ML deposition time$\;t_{ML}$ alone does not enter the equations but can be used to express all the results in the real time scale when required.

Without As desorption (at $\theta_{v}\to \infty$), we simply have $x_{g}+x_{r}=1$, because the sum of the ML growth and refill times must equal the time of ML deposition. Solutions to Equation (2) in this case are obtained in the form
(6)$$\theta =\left(\theta_{0}-\theta_{eq}-\varepsilon \right)\left(1-e^{-x/\varepsilon }\right)+x$$
(7)$$\theta_{L}=\theta_{0}-\theta +x,\;0\leq x\leq x_{g}\;\mathrm{and}\;\theta_{L}=\theta_{0}-1+x,\;x_{g}\leq x\leq 1.\;$$

The minimum As content at the end of the ML growth stage is given by $\theta_{min}=\theta_{L}\;\left(x_{g}\right)=\theta_{0}-1+x_{g}$. 

We now note that the value of ε is naturally restricted to the range $0\leq \varepsilon \leq \theta_{0}-\theta_{eq}$. According to Equation (6), the minimum ε=0 describes the instantaneous growth of fractional ML to either 1 (at $\theta_{0}-\theta_{eq}\geq 1$) or the stopping size$\;\theta_{0}-\theta_{eq}$ (at$\;\theta_{0}-\theta_{eq}<1$) [29]. The maximum$\;\varepsilon =\theta_{0}-\theta_{eq}$ describes the slowest possible growth of fractional ML at the rate of refill, θ=x, proceeding at a time-independent As content in liquid, $\theta_{L}=\theta_{0}$. Using Equation (6) at $\theta \left(x_{g}\right)=1$, the ML growth time at any ε is obtained in the form
(8)$$x_{g}=1-\left(\theta_{0}-\theta_{eq}-\varepsilon \right)+\varepsilon W\left[\frac{\left(\theta_{0}-\theta_{eq}-\varepsilon \right)}{\varepsilon }exp\left(\frac{(\theta_{0}-\theta_{eq}-\varepsilon -1}{\varepsilon }\right)\right].$$

Here, W(X) is the Lambert function such that Wexp(W)=X. Due to W(0)=0, Equation (8) gives $x_{g}=1$ at $\varepsilon =\theta_{0}-\theta_{eq}$, meaning that$\;x_{r}=0$. In this limiting case of slow island growth at the deposition rate, its growth time simply equals the ML deposition time, while the droplet at a constant content requires no refill from vapor. At ε→0, solutions to Equation (8) behave very differently depending on whether the stopping size is absent (at $\theta_{0}-\theta_{eq}>1$) or present (at$\;\theta_{0}-\theta_{eq}<1$). At$\;\theta_{0}-\theta_{eq}>1$, the main asymptote of the Lambert function is logarithmic, yielding$\;\varepsilon W\{exp\left[(\theta_{0}-\theta_{eq}-\varepsilon -1\right)/\varepsilon ]\}\to \theta_{0}-\theta_{eq}-\varepsilon -1$ and hence$\;x_{g}=0$. This solution corresponds to instantaneous ML growth without the stopping size. On the other hand, at $\theta_{0}-\theta_{eq}<1$, the Lambert function tends to zero at ε→0, giving $x_{g}=1-\left(\theta_{0}-\theta_{eq}\right)$. This solution describes the instantaneous growth of fractional ML from zero to the stopping size$\;\theta_{0}-\theta_{eq}$, which is followed by a much slower growth at the rate of refill. 

Averaging Equation (7) gives the average electron-to-hole ratio in Si-doped GaAs NWs in the form
(9)$$\frac{\langle z\rangle }{z_{0}}=\frac{\left(\theta_{eq}+\varepsilon \right)}{\theta_{0}}x_{g}+\left(1-\frac{\theta_{eq}+\varepsilon }{\theta_{0}}\right)\varepsilon \left(1-e^{-x_{g}/\varepsilon }\right)+\left(1-\frac{1}{\theta_{0}}\right)\left(1-x_{g}\right)+\frac{1}{2\theta_{0}}\left(1-x_{g}^{2}\right),$$
where $x_{g}$ is given by Equation (8). In the absence of desorption, Si doping depends on the three parameters$\;\theta_{0}$, $\theta_{eq}$ and ε. 

With As desorption, solutions to Equation (2) write
(10)$$\theta =2\beta \theta_{v}ln\left[cosh\left(\frac{\alpha x}{\theta_{v}}\right)+Csinh\left(\frac{\alpha x}{\theta_{v}}\right)\right]-\frac{\left(\theta_{eq}+\beta \theta_{v}\right)x}{\varepsilon }$$
(11)$$\theta_{L}=\alpha \theta_{v}\frac{C+tanh\left(\frac{\alpha x}{\theta_{v}}\right)}{1+Ctanh\left(\frac{\alpha x}{\theta_{v}}\right)}-\beta \theta_{v},\;0\leq x\leq x_{g},\;$$
with $\beta =\theta_{v}/\left(2\varepsilon \right)$, $\alpha =\sqrt{1+\theta_{eq}/\varepsilon +\beta^{2}}$, and $C=\left(\theta_{0}/\theta_{v}+\beta \right)/\alpha$. In this more complex case, the normalized growth time can only be obtained numerically from Equation (10) at$\;\theta \left(x_{g}\right)=1$. Then, the minimum As content in liquid $\theta_{min}=\theta_{L}\;\left(x_{g}\right)$ is calculated from Equation (11). After the ML is completed, the solution to Equation (2) for $\theta_{L}$ at dθ/dx=0 yields the result of Reference [29]: (12)$$\theta_{L}=\theta_{v}tanh\left[\frac{x-x_{g}}{\theta_{v}}+atanh\left(\frac{\theta_{min}}{\theta_{v}}\right)\right],\;x_{g}\leq x\leq x_{g}+x_{r}.\;$$

The normalized refill time is easily obtained from this equation in the form
(13)$$x_{r}=\theta_{v}\left[atanh\left(\frac{\theta_{0}}{\theta_{v}}\right)-atanh\left(\frac{\theta_{min}}{\theta_{v}}\right)\right].$$

The ML formation cycles end at $x_{T}=x_{r}+x_{g}>1.$ After some calculations, the average normalized electron-to-hole ratio can be presented as
(14)$$\frac{\langle z\rangle }{z_{0}}=\frac{1}{\left(x_{g}+x_{r}\right)}\frac{\theta_{v}^{2}}{\theta_{0}}\left\{\frac{\left(\alpha -\beta \right)x_{g}}{\theta_{v}}+ln\left[\frac{C+1-\left(C-1\right)e^{-2\alpha x_{g}/\theta_{v}}}{2}\right]+ln\left[cosh\left(\frac{x_{r}}{\theta_{v}}\right)+\frac{\theta_{min}}{\theta_{v}}sinh\left(\frac{x_{r}}{\theta_{v}}\right)\right]\right\},$$
and depends on the four parameters$\;\theta_{0}$, $\theta_{eq}$, $\theta_{v}$, and ε.

## 3. Results and Discussion

Figure 1a shows the normalized ML growth time $x_{g}\;$as a function of ε in the absence of As desorption, where the refill time $x_{r}=1-x_{g}$. The curves were obtained from Equation (8) with different $\theta_{0}-\theta_{eq}$ ranging from 0.35 to 2. In all cases, $x_{g}$ monotonically increases with ε and reaches unity at$\;\varepsilon_{max}=\theta_{0}-\theta_{eq}$, meaning that the ML growth takes a longer time for larger ε values. In the absence of the stopping effect (at$\;\theta_{0}-\theta_{eq}>1$), the curves start from zero at ε=0, which corresponds to the instantaneous growth of the whole ML. In the presence of the stopping effect (at $\theta_{0}-\theta_{eq}<1)$, a fractional ML evolves instantaneously only from zero to the stopping size$\;\theta_{0}-\theta_{eq}$ and then grows at the rate of refill, which is why$\;x_{g}$ is larger than zero at ε=0 [29]. Figure 1b shows one cycle of the ML formation in terms of the ML coverage θ and As content in liquid$\;\theta_{L}$ versus the refill x. The curves were obtained from Equations (6) to (8) at a fixed$\;\theta_{eq}=$ 0.35 and$\;\theta_{0}=$ 0.7, for two different values of ε. At a small ε= 0.02, there is a clear separation between the steps of fast growth of the ML coverage from zero to the stopping size of 0.35 ML and a much slower growth from 0.35 ML to the full ML. The As content rapidly drops to a value which is just slightly above equilibrium, and then it increases linearly to resume the initial value of 0.7 for the next nucleation event. At a large ε= 0.3, the ML grows almost at the rate of refill all the time, while the As content in liquid does not much change over the entire ML formation cycle. For the Si doping of GaAs NWs, the first case should yield a considerable drop of the electron-to hole ratio, while the second case should lead to an almost negligible effect due to the absence of concentration oscillations.

Figure 2 shows the normalized electron-to-hole ratios in Si-doped GaAs as a function of ε and$\;\theta_{0}$, along with the corresponding ML growth times, in the absence of as desorption. The curves in Figure 2a were obtained from Equations (8) and (9) at a fixed $\theta_{0}=0.7$ and two different $\theta_{eq}$=0.35 and 0.05, where the stopping effect can be present. A smaller $\theta_{eq}$ yields stronger suppression of the electron-to-hole ratio due to larger oscillations of As concentration. In both cases, the curves of $\langle z\rangle /z_{0}$ are close to$\;x_{g}$, showing that the ML growth time is the main factor determining the electron-to-hole ratio. A smaller ε value always correspond to lower $\langle z\rangle /z_{0}$. Therefore, faster island growth rates relative to refill enhance the tendency for p-type Si doping of GaAs NWs. The curves in Figure 2b were obtained from Equations (8) and (9) at a fixed ε= 0.02 (fast island growth relative to refill), and two different $\theta_{eq}\;$= 0.35 for a higher and 0.05 for a lower growth temperature. The behavior of Si doping with$\;\theta_{0\;}$is non-monotonic in both cases, revealing the tendency for achieving a minimum electron-to-hole ratio at intermediate$\;\theta_{0\;}$. At low$\;\theta_{0\;}$close to equilibrium, the VLS growth proceeds under near-equilibrium conditions, where the Si doping should be almost the same as for planar layers. Very high$\;\theta_{0\;}$are equivalent to strongly non-equilibrium VLS growth without the stopping size, where the electron-to-hole ratio is less affected by the chemical potential oscillations.

Figure 3 illustrates how the ML growth kinetics influences the Si doping process in systems with as desorption. Figure 3a shows the ML coverage and as content in liquid versus refill. The curves were obtained from Equations (10) to (13) with $\;\theta_{eq}=$ 0.05, $\;\theta_{0}=$ 0.7, and$\;\theta_{v}=$ 0.75 (high desorption rates) for two different ε values. In the regime with fast island growth (ε= 0.02), the As content rapidly drops to a minimum, which is noticeably above equilibrium. The evolution of the system in the ML growth stage is indistinguishable of that without desorption, meaning that all the arriving As atoms are consumed by the growing ML and do not re-evaporate. After the ML is completed, the refill is strongly non-linear and takes a much longer time than without desorption, so the ML formation cycle ends at $x_{T}≅1.5$. The situation is very different at a large ε= 0.5, corresponding to slow island growth relative to refill. In this case, desorption becomes important already in the stage of ML growth. This stage is much longer than without desorption, with the As concentration decreasing more slowly to a minimum that is far above the equilibrium level. On the other hand, this minimum is lower than it would be without desorption. The refill is non-linear but faster than at a small ε= 0.02. Overall, the ML formation cycle is longer ($x_{T}≅1.9$ ) than in the previous case, which is mainly due to a longer ML growth stage. 

Figure 3b shows the normalized electron-to-hole ratios in Si-doped GaAs in systems with desorption as a function of$\;\theta_{v}$. The curves were obtained from Equations (10) to (13) at a fixed ε=0.02 and$\;\theta_{eq}=$ 0.05, with (at$\;\theta_{0}=$ 0.7) and without (at $\theta_{0}=1.1$) the stopping effect. The electron-to-hole ratio gradually decreases with increasing$\;\theta_{v}$ in both cases, showing that n-type Si doping is more suppressed for lower desorption rates. Deconvolution of the total ratio $\langle z\rangle /z_{0}$ in the regime with the stopping effect shows that the contribution of the refill stage dominates over the one due to ML growth. 

Let us now formulate the most important trends for the concentration oscillations and the related Si doping of GaAs NWs in terms of temperature, NW growth rate, NW radius, and epitaxy technique. First, we have seen that larger oscillations of As concentration always lead to lower electron-to-hole ratios. The stopping effect always increases the amplitude of these oscillations, because the As concentration stays near equilibrium for a considerable fraction of the ML growth cycle. The stopping effect is observed at $\theta_{0}-\theta_{eq}<1$ and ε≪1, which requires (1) small NW radii, (2) slow NW growth rates (low$\;\theta_{0}$), and (3) low growth temperatures to avoid desorption (high$\;\theta_{v}$). Such growth regimes are more usual for the highly non-equilibrium MBE technique, where the Si doping leads to p-type conductivity [14,15,16,17,18,19,20]. Second, in the absence of the time-scale separation for the island growth and refill (large ε), we observe a new regime in which the ML develops at the rate of deposition without changing the As concentration in the droplet, and no refill is required. We suspect that such regimes occur in HVPE, which is a near-equilibrium growth process characterized by high temperatures and huge material inputs [31]. Metal organic vapor phase epitaxy (MOVPE) [32] is closer to equilibrium than MBE but yields lower NW growth rates compared to HVPE. Hence, an intermediate amplitude of the concentration oscillations is expected in MOVPE. This observation further supports the result of Ref. [22] for the n-type Si doping of HVPE-grown GaAs NWs. We strongly believe that a near-equilibrium VLS process should yield the Si doping levels in NWs similar with those in planar layers, as suggested by the curves in Figure 2b and Figure 3b. Third, the stopping effect is observed in situ only for wurtzite GaAs NWs with planar growth interface [26,27] (and growing at low rates). Zincblende NWs have the truncation at the top [24,25,26,27], which provides an additional source of material to complete the ML. This explains why the ML progression is instantaneous in truncated NWs. However, a truncated growth interface should not much affect the Si-doping process. When ML is rapidly completed by using a required amount of material from the truncation, the same amount of material must be returned to the truncation from vapor through the droplet. For Si doping, this process is equivalent to the continuing fractional ML above the stopping size at the equilibrium as concentration. Fourth, increasing $\theta_{0}$ far above the equilibrium value always suppress the concentration oscillations and hence favors n-type Si doping, as seen in Figure 2b and Figure 3b for small ε≪1. High$\;\theta_{0}$ and low$\;\theta_{eq}$ correspond to MBE growth at low temperatures and high deposition rates (which also suppresses As desorption), so the n-type Si doping of MBE grown GaAs NWs is more probable in such regimes. Finally, here, we discussed only the VLS-type Si doping through the droplets. In many cases, Si [20] and other dopants [36,37] prefer to incorporate to NWs through their side facets or even the corners separating the facets in the vapor–solid (VS) mode. For the Si doping of GaAs NWs, the VS process should lead to n-type doping and compete with p-type doping by the VLS mechanism. A competition of the VLS versus VS Si doping occurs when the NW core is grown by the VLS mode through the droplet, and the shell is grown by a step-mediated VS process on the NW sidewalls. In this case, the shell should be n-doped, while the core should be p-doped. The VS incorporation in MBE is generally preferred to the VLS one at low temperatures [3], which should again favor n-type Si doping. 

## 4. Conclusions

In conclusion, we have presented a fully analytic model for self-regulated oscillations of As concentration and the related electron-to-hole ratio in Si-doped GaAs NWs, which should be applicable in a wide range of epitaxy techniques. It has been shown that the oscillations and the Si doping process are generally controlled by the four parameters related to the maximum As concentration at nucleation, its equilibrium concentration, desorption, and the ratio of the island growth time over the deposition time. The influence of the latter parameter has never been studied before to our knowledge. However, it can completely suppress the oscillations in epitaxy techniques with very rapid NW growth rates. Generally, our study reveals the two limiting regimes of NW growth, with a large amplitude of concentration oscillations at slow NW growth rates, small NW radii, low As contents (often yielding the stopping effect), and near-equilibrium VLS processes with the slow progression of ML at an almost constant As concentration. The electron-to-hole ratio in Si-doped GaAs NWs is strongly suppressed in the first case, while in the second case, it remains the same as for the maximum as concentration at nucleation. Overall, these results should be useful for understanding and controlling the oscillatory behavior of As or P concentration in the VLS growth and the doping process in III–V NWs at low group V concentrations in the catalyst droplets.

## Figures and Tables

**Figure 1 nanomaterials-10-00833-f001:**
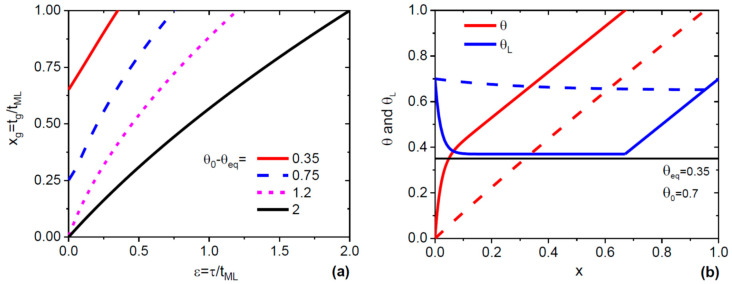
(**a**) Normalized monolayer (ML) growth time$\;x_{g}$ as a function of ε for four different values of$\;\theta_{0}-\theta_{eq}\;$given in the legend, without As desorption. At the largest$\;\theta_{0}-\theta_{eq}$ = 2, $x_{g}$ starts from zero and increases almost linearly with ε, reaching unity at$\;\varepsilon_{max}.$ The curve at $\theta_{0}-\theta_{eq}\;$ = 1.2 is more non-linear but follows the same trend. Due to the stopping effect,$\;x_{g}$ is nonzero at$\;\theta_{0}-\theta_{eq}<1$. (**b**) One oscillation of the ML coverage and As content in liquid at ε= 0.02 (solid lines) and ε= 0.3 (dashed lines). Different time scales of the fast and slow growth stages are clearly seen at ε= 0.02, where$\;\theta_{L}$ stays just slightly above equilibrium ($\theta_{eq}=$ 0.35 is indicated by the horizontal line) for a long time. At ε= 0.3, the ML growth rate is very close to the rate of refill, while$\;\theta_{L}\;$remains almost constant over the ML formation cycle.

**Figure 2 nanomaterials-10-00833-f002:**
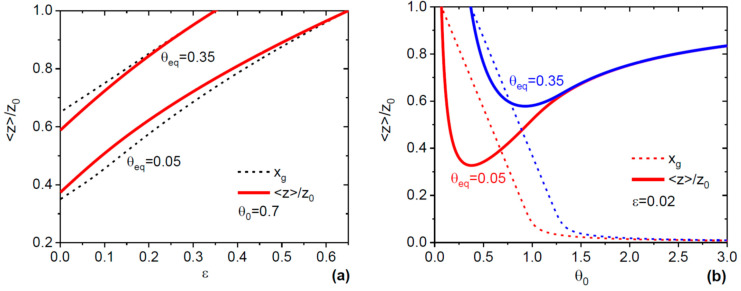
(**a**) Normalized electron-to-hole ratio in Si-doped GaAs nanowires (NWs) versus ε (solid lines) at a fixed $\theta_{0}=0.7$ and two different $\theta_{eq}$= 0.35 and 0.05, corresponding to a higher and lower temperature, respectively. Short-dashed lines show the$\;x_{g}$ values for the same parameters. The electron-to-hole ratio follows the same trend as the ML growth time, increasing quasi-linearly with ε from a minimum for instantaneous island growth to unity at$\;\varepsilon_{max}=\theta_{0}-\theta_{eq}$. (**b**) Same as (**a**) but at a fixed ε= 0.02 as a function of $\theta_{0}$. The values of $\langle z\rangle /z_{0}$ tend to unity when$\;\theta_{0}$ is close to equilibrium (near-equilibrium growth conditions) and at large$\;\theta_{0}$ (strongly non-equilibrium growth conditions), and they reach their minimum at intermediate$\;\theta_{0}$.

**Figure 3 nanomaterials-10-00833-f003:**
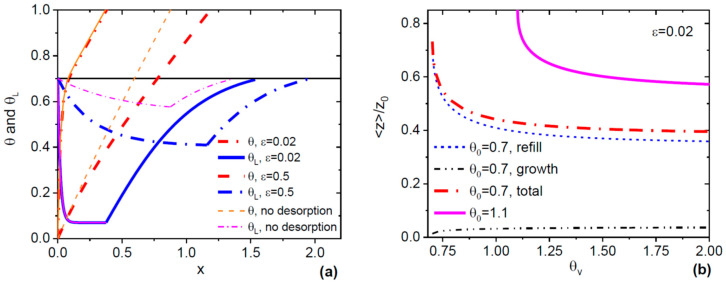
(**a**) One oscillation of the monolayer (ML) coverage and As content in liquid affected by As desorption, at$\;\theta_{eq}=$ 0.05, $\;\theta_{0}=$ 0.7, $\;\theta_{v}=$0.75, and two different ε= 0.02 (bold solid lines) and 0.5 (bold dashed lines). Thin lines show the corresponding behavior which would be observed without As desorption in the ML growth stage. At small ε, the growth stage is not affected by desorption, while the refill stage is prolongated. The As content drops very sharply as the ML grows. At large ε, the ML growth stage is much longer than without desorption and contributes to the total prolongation of the ML formation cycle. The As concentration varies much less than at small ε, as we saw earlier in Figure 1b. (**b**) Normalized electron-to-hole ratio in Si-doped GaAs NWs in systems with As desorption at a fixed ε= 0.02 versus$\;\theta_{v}$, at two different$\;\theta_{0}=$ 0.7 and 1.1 (bold lines). At$\;\theta_{0}=$ 0.7, the dash-dotted and short-dashed lines show the contributions of the ML growth and refill stages into the resulting ratio $\langle z\rangle /z_{0}$. The contribution of refill is dominant for these parameters. At$\;\theta_{0}=1.1$, the ML growth is instantaneous, and the decrease of the electron-to-hole ratio is only due to refill. Overall, a smaller$\;\theta_{0}$ yields a stronger suppression of n-type Si doping. The ratio $\langle z\rangle /z_{0}\;$monotonically decreases with increasing$\;\theta_{v}$, meaning that higher As desorption rates lead to larger electron-to-hole ratios.

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
