# Peer review of "Oscillations of As Concentration and Electron-to-Hole Ratio in Si-Doped GaAs Nanowires"

_nanomaterials, 2020, doi:10.3390/nano10050833_

Round 1
Reviewer 1 Report
The manuscript "Oscillations of As concentration and electron-to-hole 2 ratio in Si-doped GaAs nanowires" on the theoretical investigation of factors influencing the As depletion in the vapor-liquid-solid grown Si-doped GaAs nanowires.
From a general point of view the topic of the manuscript is interesting and worth of investigation.
The introduction clearly states the aim of the work and sharply inserts the work within a general scientific and technological framework of interest. The theoretical approaches and methods are clearly described and strongly founded. Results are quite interesting and potentially useful. The discussions of the results are clear and reliable allowing to draw some new and interesting insights.
In addition, the manuscript is generally clear, well-written and well-organized. Figures are clear and appealing.
Overall, I find a valuable and interesting manuscript which deserves publication as it is.
Reviewer 2 Report
In the manuscript „Oscillations of As concentration and electron-to-hole 2 ratio in Si-doped GaAs nanowires” by V. Dubrovskii et al. the authors report on a fully analytic model for self-regulated oscillations of As concentration and the related electron-to-hole ratio in Si-doped GaAs NWs, which should be applicable in a wide range of epitaxy techniques. They demonstrate that the oscillations and the Si doping process are generally controlled by the maximum As concentration at nucleation, its equilibrium concentration, desorption, and the ratio of the island growth time over the deposition time. These results should be useful for understanding and controlling the oscillatory behavior of As concentration in the VLS growth and the doping process in III-V NWs and is, therefore, highly relevant for the scientific community.
The manuscript is in general well written and easy to understand. I believe that the manuscript could be considered for publication after some comments were addressed.
- Page 2 line 61: typo: at very low instead of al very low.
- How will the conclusions of this work be affected when growing core-shell NWs where the core is grown by VLS growth, but the shell is grown by a step mediated process?
- In the introduction the authors could also mention works on InP/InAsP photodetectors (Nano Letters 17, 3356 (2017) and GaNP NWs for light emission (Nano Letters 15, 242 (2015).
Reviewer 3 Report
GaAs NWs play a leading role within the III-V semiconductor NW family due to their unique optical/photonic and optoelectronic properties. In this system, accurate design and control of dopant incorporation mechanisms and of doping levels represent a vital requirement for engineering task-specific structure-property relationships. In this context, the work by Dubrovskii and Hijazi shines light on the impact of the self-oscillations displayed by group V concentration in a catalyst droplet on the e/h ratio, assuming as material platform Si-doped GaAs NWs grown by VLS method. Together with self-oscillations, the authors target the potential impact of the stopping effect at low As concentration and of the maximum As concentration at nucleation. Overall, the e/h ratio is mapped across different self-oscillation regimes, yielding to different phenomenologies. Noticeably, the analytical model proposed and discussed to investigate the e/h ratio is general and thus the main theoretical findings are expected to apply to GaAs NW-based systems grown by any epitaxial technique. The references encompass however experimental works (reporting optical/electrical studies) on doped GaAs NWs (or closely related systems) grown basically by MBE. See for instance ref. 7,8,10,14, 19 among others. More importantly, the authors claim in the conclusions that the results reported in the paper “should be useful for understanding and controlling the oscillatory behavior of As concentration in the VLS growth and the doping process in III-V NWs”. Do similar processes (As or other group V element concentration oscillations during growth) occur frequently across III-V NWs? May a counterpart of this process exist for GaAs NWs grown with different epitaxial techniques (e.g. CBE) and different dopants (e.g. Se)?
Beyond these general considerations, this is a very technical and very solid piece of work that addresses a topic of undoubted relevance in the field of bottom up III-V semiconductor nanostructures at large: I heartily recommend its publication, and I believe this is a brilliant example of that kind of paper that may contribute to widening the readership of Nanomaterials, by catching the attention of the semiconductor NW community working on theory and modeling.
While going through the manuscript in Fig. 3 I was a bit confused about the identification of curves and related conditions: the choice of colors can be misleading. If one prints the manuscript in grey color scale, then all figures become quite obscure, indeed. Maybe the authors could find better codes to label each curve in the figures.
